# The Pathogenesis of End-Stage Renal Disease from the Standpoint of the Theory of General Pathological Processes of Inflammation

**DOI:** 10.3390/ijms222111453

**Published:** 2021-10-23

**Authors:** Evgenii Gusev, Liliya Solomatina, Yulia Zhuravleva, Alexey Sarapultsev

**Affiliations:** Institute of Immunology and Physiology, Ural Branch of the Russian Academy of Sciences, 620049 Ekaterinburg, Russia; gusev36@mail.ru (E.G.); jazhur@mail.ru (Y.Z.); asarapultsev@gmail.com (A.S.)

**Keywords:** cellular stress, chronic low-grade inflammation, chronic systemic inflammation, classical inflammation, end-stage renal disease, cytokines

## Abstract

Chronic kidney disease can progress to end-stage chronic renal disease (ESRD), which requires the use of replacement therapy (dialysis or kidney transplant) in life-threatening conditions. In ESRD, irreversible changes in the kidneys are associated with systemic changes of proinflammatory nature and dysfunctions of internal organs, skeletal muscles, and integumentary tissues. The common components of ESRD pathogenesis, regardless of the initial nosology, are (1) local (in the kidneys) and systemic chronic low-grade inflammation (ChLGI) as a risk factor for diabetic kidney disease and its progression to ESRD, (2) inflammation of the classical type characteristic of primary and secondary autoimmune glomerulonephritis and infectious recurrent pyelonephritis, as well as immune reactions in kidney allograft rejection, and (3) chronic systemic inflammation (ChSI), pathogenetically characterized by latent microcirculatory disorders and manifestations of paracoagulation. The development of ChSI is closely associated with programmed hemodialysis in ESRD, as well as with the systemic autoimmune process. Consideration of ESRD pathogenesis from the standpoint of the theory of general pathological processes opens up the scope not only for particular but also for universal approaches to conducting pathogenetic therapies and diagnosing and predicting systemic complications in severe nephropathies.

## 1. Introduction

Currently, it is difficult to find a disease the pathogenesis of which researchers do not come across certain molecular mechanisms of inflammation. As a result, the definition of the term “inflammation” in many studies goes much beyond the classical concepts of this broad pathological process. At the same time, it is often unclear which processes these identified molecular mechanisms relate to. This concerns both the accumulation of inflammatory mediators in the blood and the presence of other signs of a systemic inflammatory response (SIR). These signs have been detected to range from extreme physiological processes to critical shock conditions. That is why “inflammation” requires classification into several independent common pathological processes to determine their common and different molecular and cellular mechanisms. From these vantage points, a large group of diseases is a convenient material for describing the role of key general pathological processes associated with inflammation. These diseases are united by the general concept of “chronic kidney disease (CKD)” with a possible progressive course to the terminal stage of its development.

We think this article could be interesting not only for nephrologists but also specialists in related clinical disciplines, as well as pathological physiologists, general pathologists, immunologists, and other fundamental medicine and molecular biology representatives. With this in mind, we outline the causes and consequences of end-stage renal disease (ESRD) in general. We have included a brief description of the general pathological processes under consideration, so that the article’s readers can understand the starting points of our analysis and the conclusions we reached.

This review reveals the general and distinctive molecular and cellular mechanisms of various general pathological processes of inflammation, as well as their interconnections during the development of severe renal pathology.

## 2. General Characteristics of the Causes and Consequences of ESRD

CKD appears to be an integral pathology with heterogeneous etiological and pathogenetic factors of onset and development. A common characteristic of CKD is the presence of chronic morphological and functional disorders in the kidneys, leading to renal failure. The prevalence of CKD among the world’s population currently ranges from 11% to 13% [1,2]. The progressive course of CKD is associated with a steady decline in the glomerular filtration rate to ESRD. The fifth stage of CKD, ESRD, necessitates renal replacement therapy, such as peritoneal or programmed hemodialysis (PH; “artificial kidney”), or donor kidney transplantation. However, allograft transplantation is available only for 10% of patients requiring new organs [3]. The prevalence of ESRD is a serious social problem in many countries around the world. Currently, approximately three million ESRD patients worldwide are constantly dependent on PH, which only partially replaces kidney function. The number of people with ESRD in the United States alone exceeded 785,000 by the end of 2018 [4], and there is a steady upward trend in the detection of this pathology [5]. This is because, during hemodialysis, mainly small solutes are removed from the bloodstream, while larger protein-bound uremic toxins remain in the blood [2]. The complex action of ESRD-related pathogenic factors causes arterial sclerosis and calcification, as well as other alterations in the cardiovascular system [6]. In ESRD, disturbances in this system can be induced by nephrotoxins that cause direct injury to the heart and blood vessels, plus secondary metabolic disorders and neuroendocrine dysfunctions [7].

A diagnosis of CKD is made when the structure and function of the kidneys have been impaired for more than 3 months. The degree of proteinuria and the glomerular filtration rate (GFR) are the two most important indicators of the CKD stage. The fifth, terminal, stage of CKD is defined by a GFR of less than 15 mL/min [1,5]. However, clinical signs of nephrotic syndrome, which could potentially exacerbate the progression of renal failure (edema, proteinuria >3.5 g/day, hypoalbuminemia, and hypercholesterolemia), can appear in the early stages of CKD [5,8]. Uremic toxicity is an indication for the initiation of renal replacement therapy and can manifest itself in the form of anorexia, nausea, vomiting, bleeding diathesis, pericarditis, uremic neuropathy or encephalopathy, seizures, coma, and death [9,10]. Advanced age, smoking, obesity, and other unhealthy lifestyle factors [5,11,12,13], along with genetic susceptibility [14,15], are all risk factors for the development of CKD preceding ESRD.

Diabetes mellitus, hypertension, primary and secondary (caused by systemic lupus erythematosus, systemic vasculitis, myelo- and lymphoproliferative, and other diseases) glomerulonephritis, polycystic kidney disease, obstructive uropathy, vesicoureteral reflux, renal amyloidosis, and drug nephropathy are the most common causes of ESRD in the developed countries [5,10,11,16]. Chronic pyelonephritis is an inflammatory disease of the pyelocaliceal system involving tubulointerstitial tissue, which leads to tubulointerstitial fibrosis and, eventually, complete nephrosclerosis if it is not treated. Chronic pyelonephritis, along with other infections that cause the development of various glomerular and tubulointerstitial renal diseases, makes a relatively more significant specific contribution to the development of ESRD in patients from developing countries [17]. Autosomal dominant tubulointerstitial kidney disease includes rare kidney diseases characterized by tubular damage and interstitial fibrosis in the absence of glomerular lesions, with inevitable progression to ESRD [18].

The development of CKD and its progression to ESRD remain significant factors in the decline in quality of life and premature death. Mortality in ESRD patients is significantly higher than in CKD patients without ESRD, and, even with the modern techniques of hemodialysis, the death rates vary from 20% to 50% over 24 months [5]. According to an autopsy study, the main causes of death of patients with ESRD are pathology of the cardiovascular system, followed by infections, as well as cerebrovascular, metabolic, and other causes [11,19]. Furthermore, the initial (pre-trauma) ESRD is an independent risk factor for death in critical cases of various genesis (for example, in acute trauma) [20]. The acquired (secondary) cystic disease is frequently associated with ESRD and is a risk factor for renal cell carcinoma [21]. The most common morphological forms of glomerulonephritis leading to ESRD, as well as glomerulonephritis leading to death in ESRD, are usually associated with (a) focal segmental glomerulosclerosis, (b) membranous glomerulonephritis, (c) mesangial proliferative glomerulonephritis, (d) immunoglobulin A (IgA) nephropathy (mesangial and endocapillary hypercellularity >50% glomeruli, glomerulosclerosis, tubular atrophy, and interstitial fibrosis), (e) lupus nephritis, and (f) kidney damage in Shenlein–Henoch purpura (focal segmental proliferative glomerulonephritis and rapidly progressive crescentic glomerulonephritis) [22,23,24].

Renal fibrosis is a typical final stage of inflammation that occurs in nearly all nephropathies [25]. Fibrosis can affect all compartments of the kidney, ultimately causing destruction of the renal parenchyma and leading to ESRD [26]. Local and systemic disruptions of oxygen transport with the formation of hypoxia are common causes of renal sclerosis [27].

Many systemic factors of progressive renal failure aggravate it via the mechanisms of vicious pathogenetic cycles [5,10,11,28,29,30] and include the following:poisoning of the body with nephrotoxins (various middle molecules, derivatives of phenol and indole, homocysteine, and other molecules);excess in the blood of potentially toxic water-soluble drugs;hypoproteinemia, hyperlipidemia, hyperphosphatemia, hyperkalemia, hyponatremia, hyperuricemia, and metabolic acidosis;hypertension, accelerated development of atherosclerosis and heart failure, and rapid progression of diabetes mellitus (with diabetic kidney disease—DKD);anemia (decreased production of erythropoietin and iron absorption [31,32]);thrombophilia and thrombocytopathy;systemic proinflammatory processes;dysfunction of the renin–angiotensin–aldosterone system, as well as other neuroendocrine dysfunctions.

Various indications of immunological dysfunction, leading to infectious complications [33], including accelerated aging of the thymus and other lymphoid organs and insufficient adaptive immune response [34,35], characterize uremia at the systemic level. An increase in systemic proinflammatory processes is also a hallmark of this disorder, which is linked to premature body aging [36]. Disorders of both renal and systemic microcirculation, as well as a substantial decrease in microvessel density, are common abnormalities in the course of CKD [37]. As a result of these alterations in the context of ESRD, other organs begin to fail, including cardiovascular disease progression, cerebrovascular pathology, muscle atrophy, and cachexia [37]. Pathological alterations in the endothelium (endotheliosis) are the primary source of systemic microvascular pathology in ESRD, necessitating general treatment strategies employed in many cardiovascular and renal illnesses [38]. Metabolic abnormalities in ESRD demand the same treatment concepts as other diseases, such as statin use [39].

The mutual negative influence of impaired renal function and cardiovascular disease can lead to cardiorenal syndrome [40,41,42]. Morbid obesity and metabolic syndrome, which can lead to type 2 diabetes mellitus, are common satellites of cardiorenal syndrome in this situation [13,43]. In chronic renal failure, abnormalities in other organs, such as the lungs [44,45] and the brain [46], are also seen as contributing to persistent dysfunction of those organs. Moreover, even after accounting for the effects of age, diabetes, and depression, individuals with ESRD-associated DKD have an accelerated brain aging phenotype [47]. As a result, substantial injury to some organs, such as the liver in hepatorenal syndrome, can result in secondary impairment of renal function, eventually leading to severe renal failure [48,49,50].

There is no doubt that pattern recognition receptors (PRRs) play a key pathogenetic role in ESRD-associated local and systemic disturbances [51]. Their ligands are conservative microbial structures—pathogen-associated molecular patterns (PAMP) and damage-associated molecular patterns (DAMP). They are capable of activating the immune system cells, as well as endotheliocytes and platelets, causing proinflammatory cellular stress in them [52]. The DAMP category includes not only many tissue destruction products but also some uremic toxins, including uric acid [53]. The accumulation of elevated PAMP concentrations in the blood in ESRD patients may, in turn, be associated not only with infection but also with impaired microbiota and intestinal barrier function for PAMP translocation [54,55,56]. At the same time, local and systemic manifestations of inflammatory reactions are typical causes for the development of CKD and its progression to ESRD. Currently, there are three “large” general pathological processes associated with inflammation: (1) systemic/local chronic low-grade inflammation (ChLGI); (2) local and systemic manifestations of inflammation of the classical (canonical) type; (3) systemic inflammation, which is critical for the life of patients [52]. Distinctive features of these processes are different relations toward the severity and prevalence of proinflammatory tissue stress and the associated phenomenon of inflammatory microcirculation of blood (Figure 1). Next, we dwell on the role of these processes in the pathogenesis of ESRD in more detail.

## 3. Cellular Stress as a Common Pathogenetic Basis for General Pathological Processes Associated with Inflammation

In pathogenic processes, cellular stress is a fundamental functional unit. It is found in all types of cells in some form or another, but it is more prevalent in the immune system, which requires proinflammatory stress to execute its primary activities. In this case, the following universal and interrelated components of cellular stress can be distinguished [52,57]:oxidative stress;cell response to DNA damage;mitochondrial stress, including mitochondrial unfolded protein response (UPRmt);the stress of the endoplasmic reticulum (ER), including calcium-dependent mechanisms and UPRER;response of inducible heat-shock proteins (HSPs), including their participation in the UPR;inhibition (during cell growth) or intensification of autophagy processes (utilization of altered organelles and macromolecules) and other manifestations of lysosomal stress;inflammasome formation;formation of stress noncoding microRNAs (miRNAs);formation of an intracellular network of signaling pathways of cellular stress;formation of proinflammatory receptor and secretory cell phenotype.

At the cell level, the development of stress is mediated by complex programs of epigenetic control and the interweaving of various inducible signaling pathways, the elements (regulatory proteins) of which are constantly subjected to multiple post-translational modifications [58]. At the same time, various extracellular and intracellular stress signals can activate in different cells common collector type protein kinases (for example, MARK, Akt, PI3K, PKC, ATM, ATR, AMPK, PKA, PKR, and mTOR) and universal transcription factors of cellular stress (for example, NF-κB, p53, AP-1, HIF, HSF, Nrf2, and ATF4). Although the same signaling molecules can be activated in a variety of ways and participate in a variety of processes, typical patterns of their interaction can be identified. Thus, the signaling pathways p53 and NF-κB can competitively inhibit each other [59,60]. For example, at a relatively moderate level of oxidative stress, NF-κB is not activated, but one observes p53-mediated DNA repair or apoptosis of irreversibly damaged proliferating cells. A further increase in the level of oxidative stress activates NF-κB and inhibits p53-induced cell apoptosis, which contributes to the resistance of cells to oxidative stress and an increase in their proinflammatory activity [59,61]. In particular, NF-κB promotes the formation of a pronounced proinflammatory phenotype in macrophages and inhibits their proliferation, but enhances differentiation [62]. Key roles in the development of cellular stress are played by the following [52,63]:the transcription factor HIF-1 (hypoxia-inducible factor-1) during hypoxia;the HSF1 factor for HSP production;the Nrf2 factor (to trigger the production of antioxidants through a negative feedback mechanism) in case of oxidative stress;ATF4 plays, along with HSF, an essential role in the development of UPRmt and UPRER.

The emergence of a proinflammatory phenotype in a large number of cells at once determines the effect of their network interaction with the development of tissue stress, for example, through the formation of a cytokine network. Tissue proinflammatory stress manifests itself in a variety of ways, all of which are fundamentally different. At the same time, the same signaling pathways are found in cells involved in various types of inflammation and para-inflammation. These processes involve various types of activated cells, with the defining role pertaining to cells of the immune system.

## 4. Typical Low-Intensity Inflammatory Processes in CKD and ESRD

### 4.1. General Patterns of Development of Chronic Low-Grade Inflammation (ChLGI)

Characteristic features of ChLGI are the following [52]:ChLGI is tissue stress in response to local or systemic damage, which is insufficient for the development of classical or systemic inflammation, respectively.ChLGI is characterized by relatively low manifestations of SIR: an increase in C-reactive protein (CRP) in the blood, usually within the marginal zone between norm and pathology (3-10 mg/L), and an increase in key proinflammatory cytokines up to 2–4 times higher than the upper limit normal value.Signs of tissue decay and systemic coagulopathy are not typical; signs of organ dysfunction develop slowly; the accelerated development of atherosclerosis, hypertension, and tissue aging is characteristic; there is no connection of these changes with systemic manifestations of infections and autoimmune diseases, with pronounced signs of chronic classical inflammation.Key inducers of ChLGI are metabolic factors with a low ability to induce tissue changes, including modified proteins (denatured, oxidized, glycated), high concentrations of saturated free fatty acids (FFA), and oxidized low-density lipoproteins (oxLDL), homocysteine, and many other metabolites. The gradual accumulation of damage to the genome, proteome, and metabolome, as well as dysfunctions of organs during aging, contribute to an increase in the proinflammatory status of the organism.ChLGI involves a large number of parenchymal and stromal cells of various organs with relatively weak participation of “professional cells” of inflammation (leukocytes and their descendants, characteristic of the focus of inflammation). Consequently, ChLGI lacks barrier functions and many other signs of classical inflammation, including hyperemia, edema (exudation), and pronounced leukocyte infiltration, which determines the basic cellular composition of the inflammation focus [64,65].ChLGI can be defined as a para-inflammatory or quasi-inflammatory process. However, some of its features can be found in physiological processes. For example, in healthy people, the reaction of the acute phase of the liver can be detected situationally [66], and the physiological state of the intestinal mucosa is characterized by the presence of inflammasomes in the epithelial cells that perform a protective function against infection and tumors [67]. In athletes with pronounced work of skeletal muscles, there may be a significant increase in the blood levels of proinflammatory cytokines, especially IL-6, and other proinflammatory myokines [68].Pathological systemic manifestations of ChLGI can be directly associated with metabolic syndrome and, especially, with type 2 diabetes mellitus, as well as with neurodegeneration and chronic heart failure in the elderly [52]. Currently, it is very difficult to separate systemic ChLGI from SIR of classical inflammation in autoimmune, infectious, and many other chronic diseases due to the functional overlap of these processes (Figure 1).

The same proinflammatory cellular stress signaling pathways linked to tissue development can be observed in both health and disease [69]. In particular, the PI3K-1/Akt/mTOR signaling pathway associated with tissue growth is triggered by insulin, many cytokines and growth factors, DAMP and PAMP, and some adipokines (lipokines) in many cell types [70]. ATP deficiency or further enhancement of oxidative stress induces AMPK (5′ AMP-activated protein kinase) signaling pathways, which also leads to the development of cellular stress, activation of lipolysis (in adipocytes), proteolysis, and autophagy through inhibition of the mTOR pathway [71]. Thus, cellular stress can include both competitive, alternative, signaling pathways and features of the cellular phenotype, as well as general molecular mechanisms for various variants of its development.

To prevent the uncontrolled escalation of cellular stress, negative feedback mechanisms are activated [52]. In particular, essential omega-3 polyunsaturated fatty-acid derivatives (lipoxins, resolvins, protectins, and maresins) act as functional antagonists of proinflammatory eicosanoids while enhancing the clearance of pathogens [72,73]. Over a 25 year follow-up in the United States, the amount of dietary omega-3 polyunsaturated fatty-acid intake was inversely proportional to the incidence of CKD [74]. Some lipokines (adipokines), for example, adiponectin and omentin-1, whose production is activated in obesity and type 2 diabetes mellitus, have anti-inflammatory activity [75]. Almost any mechanism of cellular stress development includes a link of negative feedback, limiting its development and making it potentially reversible [52].

A large group (~30) of scavenger receptors (SR) play a key role in the uptake of aberrant cells and metabolites by macrophages and some other cells, including oxLDL and glycation end products (AGEs) [76]. These receptors functioning at the junction of immunity and metabolism, as well as norm and pathologies, are involved in the regulation of cellular and tissue stress and in CKD and ESRD pathogenesis [77,78,79].

Normally, the physiological processes of para-inflammation are limited in time, space, and intensity. However, the processes of para-inflammation can acquire a pathological character and contribute to a stable change in homeostasis—allostasis [80]. Increased levels of circulating cytokines and other proinflammatory agents, such as adipose tissue lipokines, are a key pathophysiological link between health, chronic disorders of the internal organs, and aging [52,81,82]. The presence of a systemic proinflammatory status, for example, in morbid obesity, is a risk factor for exercise intolerance, development of microvascular dementia, coronary microvascular angina pectoris, heart failure, chronic obstructive pulmonary disease, and CKD [83]. These changes are associated with endothelial dysfunction, which, in turn, affects the control of vascular tone by disrupting the secretion of paracrine factors, including a decrease in nitric oxide (NO) production and an increase in endothelin-1 (ET-1) production, contributing to hypertension [83].

In general, the escalation of systemic ChLGI clinically manifests itself in the development of metabolic syndrome, type 2 diabetes mellitus, atherosclerosis, sarcopenia, hypertension, neurodegeneration, atrophic and sclerotic changes in internal organs, immune dysfunction, and a decrease in functional reserves of organ systems [52,76,84]. The general laws of these processes are the intensification of the processes of cell apoptosis, gradual tissue atrophy, and tissue sclerosis. In the kidneys, these processes can contribute to a more intense local manifestation of ChLGI with the development of DKD, also known as diabetic nephropathy.

### 4.2. Pathogenetic Significance of ChLGI in the Onset of ESRD

Currently, the most convincing evidence for the role of ChLGI in the development of CKD to ESRD comes from DKD. This pathology is characterized by both hemodynamic (hyperfiltration) and structural abnormalities (glomerulosclerosis, interstitial fibrosis) (Figure 2). AGEs and the main receptor for their binding, namely, SR-J1 (RAGE), can play a significant role in the pathological activation of mesangial macrophages and podocytes, and glycation of basement membrane collagen is one of the key mechanisms of dysfunction of glomerular capillaries and adjacent podocytes [77]. The development of nephropathy is also facilitated by ectopic lipid deposition in the kidneys, excess FFA, and the activation of the scavenger receptor SR-B2 (CD36) [79]. A wide range of ligands, including AGE, FFA, and oxLDL, activate this receptor [76]. As you know, the insulin-dependent transporter, GLUT4, does not mediate the transport of glucose into endothelial cells. Therefore, in diabetes and hyperglycemia inside these cells, the accumulation of glucose, sorbitol, and other glucose metabolites is also a factor of endotheliosis [85].

Even with moderate but stable hyperglycemia in type 2 diabetes mellitus, there may be a pathological accumulation in the blood of AGE and other aberrant metabolites, some DAMPs, including HMGB1 (high-mobility group protein B1), and an increase in the blood of proinflammatory cytokines (tumor necrosis factor-alpha (TNF-α), IL-1β, and IL-6), which is accompanied by proinflammatory activation of mesangial macrophages and other cells [86]. At the same time, with rapidly progressing DKD in the blood, there is a significant accumulation of the following [86,87]:proinflammatory cytokine of activated macrophages IL-18 (also accumulates in the urine);soluble forms of TNF receptors (sTNFR1 and sTNFR2);soluble forms of endothelial adhesion receptors (sICAM-1/CD54 and sVCAM-1/CD106);reactive oxygen species (ROS), oxLDL, AGE, and other biomarkers of systemic ChLGI.

SIR factors, in turn, are involved in the disruption of various metabolic processes. In particular, IL-6 disrupts iron metabolism, which is an additional mechanism of anemia in ESRD [88]. Against the background of systemic ChLGI, many aberrant metabolites, proinflammatory cytokines, and cytokine-like factors of extrarenal tissues can influence the development of CKD. Specifically, many lipokines (adipokines) of adipose tissue can be attributed to these factors in type 2 diabetes mellitus. At the same time, the level of adiponectin in the blood can be a good predictor of DKD [89], and resistin-critical complications of CKD of various origins [90].

These systemic changes are characteristic of both type 1 and type 2 diabetes mellitus. However, in type 2 diabetes mellitus, structural abnormalities in the kidneys are more heterogeneous and less correlated with clinical manifestations [91]. Overall, about 30–40% of diabetic patients develop nephropathy, and this kidney damage usually progresses in about one-third of patients [92]. The conditions for the development of DKD are many risk factors, including genetic predisposition from inducible genes of cellular stress, for example, polymorphism of genes encoding HSP70 [93].

Changes in the renal corpuscles, such as the thickening of the basement membranes of the renal glomeruli, mesangial expansion, and hypertrophy along with loss of podocytes, sclerosis of a greater number of glomeruli, and exudative damage with the development of mesangiolysis, are typical morphological manifestations of DKD (fibrin deposition) [77,94]. Interstitial fibrosis and tubular atrophy in the renal medulla, hyalinosis of arterioles and arteriosclerosis, expansion of the subendothelial space, and neovascularization can also be revealed [94]. These changes have individual signs characteristic of the focus of classical inflammation (for example, interstitial fibrinous edema) and, in some cases, are accompanied by the involvement of migrating leukocytes, including T and B lymphocytes, in the process [92,94,95,96]. As a result, the pathogenesis of DKD is characterized not only by metabolic abnormalities but also by multiple immunological and inflammatory mechanisms [96,97], making it a challenge to distinguish between ChLGI and traditional inflammation processes in this pathology.

The characteristic typical signs of renal fibrosis are the development of cellular and tissue proinflammatory stress in the kidneys, including oxidative stress, the formation of inflammasomes, and factors of the proinflammatory secretory phenotype [98,99]. Thus, regardless of the precise causal mechanism or localization of the process, the universal effect of cytokines and other mediators of various renal cells on the development of renal fibrosis has been demonstrated in people and animals [100,101,102,103,104,105], as listed below.
angiotensin II (Ang-II), ET-1, adenosine (ADO), ROS;transforming growth factor-β (TGF-β);growth factors of platelets (PDGF), connective tissue (CTGF), and fibroblasts-23 (FGF-23);chemokines: monocyte chemoattractant protein-1 (MCP-1, CCL2) and stromal cell factor-1 (SDF-1, CXCL12);cytokines: TNF-α, IL-6, IL-11, IL-18, and IL-20;kidney damage molecule-1 (KIM-1);direct intercellular contact interactions of Notch family receptors and Notch ligands involved in the epithelial–mesenchymal transition.

Several of these factors can act as renal toxins. At the same time, an increase in the blood metabolic cytokine, FGF-23, which is involved in the regulation of circulating phosphate and vitamin D levels, is a reliable prognostic marker for assessing the risk of ESRD and unfavorable outcomes of this complication [106,107].

Even before the appearance of obvious signs of classical inflammation, the involvement of mast cells may be noted in the process of renal fibrosis in reflux nephropathy [108] and renal amyloidosis [109]. In many respects, cellular apoptosis, parenchymal dystrophy, and renal fibrosis are associated with hemodynamic disturbances due to decreased saturation and hypoxia, which initiate the following [103]:oxidative stress;many signaling pathways of cellular stress, including Notch, MARK, PKC, PI3K/Akt/mTOR, and AMPK;activation of key factors of cellular stress transcription: NF-κB, AP-1, and HIF;the action of many regulatory miRNAs.

In general, the development of renal fibrosis and ESRD depends on the degree of systemic and local homeostatic and proinflammatory changes in the kidneys, alongside dysfunctions from other organs, for example, in cardiorenal syndrome [40,41,42].

The importance of local manifestations of ChLGI in the formation of both autosomal dominant and autosomal recessive polycystic kidney disease has also been proven [110]. Interstitial inflammation is linked to the formation of cysts, as well as fibrosis, in these cases. Furthermore, indicators of inflammation and macrophage infiltration of the renal interstitium appear much before cyst formation is identified, and they are linked to disease progression.

It is important to remember that DKD creates a vicious pathogenetic cycle since chronic kidney disease (CKD) contributes to the development of systemic ChLGI, insulin resistance, and the formation of metabolic syndrome in people who do not have it initially [111]. To begin with, the development of systemic proinflammatory tissue stress is promoted by the presence of ESRD and PH due to multiple sources of inflammatory stimuli, such as oxidative stress, acidosis, and fluid overload, increased comorbidity, including infections, dialysis procedure, production, and inadequate removal (insufficient clearance) of proinflammatory cytokines [112]. These changes in PH are associated with insufficiently effective filtration of cytokines and other molecules weighing 15–45 kDa, activation of leukocytes and plasma systems of hemostasis, complement, and kallikrein-kinins during the interaction of blood with dialysis membranes, cyclical fluctuations in blood volume and electrolyte levels, and osmotic shifts [113,114,115]. Uremia, as a common link in the pathophysiology of ESRD, causes tissue stress in adipose tissue and skeletal muscles, increasing insulin resistance and leading to the development of SIR in facultatively glycosylating organs [116].

Thus, against the backdrop of systemic ChLGI, the progression of CKD to ESRD is the result of a complex interplay of local and systemic genetic, ontogenetic, and environmental variables, as well as metabolic and immunological components. As a result, ESRD hastens the emergence of systemic processes that are unfavorable to patients. The variety and proportion of proinflammatory mechanisms determine the pathogenesis of specific CKD variants.

## 5. Typical Patterns of Classical Inflammation in CKD and ESRD

### 5.1. General Patterns of Classic Inflammation

The presence of a microcirculatory network in vertebrates, which determines the possibility of exudative–vascular reactions and directed transendothelial migration of leukocytes, as well as the presence of a system of lymphoid organs, responsible for the development of adaptive immunity [64,117], makes classical inflammation unique to them. As previously stated, the formation of a focus, which limits the affected area, is the fundamental sign of canonical inflammation. Migrating to the inflammation focus leukocytes and macrophages, which are formed from monocytes, perform the determining or significant (in the case of chronicity) proinflammatory function in inflammation focus.

According to the predominance of one or another pathogenetic mechanism and the cellular composition of classical inflammation, the following types can be distinguished:exudative–vascular catarrhal and serous inflammation (with more pronounced exudation in serous);productive (proliferative) inflammation, with the predominance of cellular infiltration of a particular composition depending on the nature of the damage factor and the stage of the process;exudative–destructive inflammation (purulent, curd, fibrinous, and gangrenous);mixed inflammation when it is impossible to clearly define its specific type.

The interplay of adaptive and innate immunity is crucial in the development of inflammation. The macrophage is the most important cell in innate immunity (M). Macrophages can activate and polarize in two main competitive directions during the development of inflammation: the classical form of activation and differentiation—M1, and the alternative—M2 [118]. These types of macrophages enter into cooperative relationships with CD4^+^ T-helper type 1 (Th1), which are key producers of interferon-gamma (IFN-γ), or with Th2 (IL-4, IL-5, and IL-13), respectively [118,119]. In addition, some M2 subpopulations cooperate with Th17 (IL-17A/F, IL-21, IL-22, and TNF-α) and T-regulatory cells—Treg (IL-10 and TGF-β) [118,119]. Differentiation of M is plastic and one can speak of their morphofunctional drift in the M1–M2 range rather than anything else [120,121]. In this case, the dynamics in the direction M2 → M1 usually indicate an increase in damage and inflammation, whereas fibrosis is indicated in the opposite direction [52]. Th differentiation is also plastic, and, under the action of cytokines and other stimuli, what can occur is the reprogramming of T-helpers from one subpopulation to another and formation of cells with a mixed phenotype and function [122,123,124,125]. Differentiation of stromal macrophages, including during the development of ChLGI, is also characterized by morphofunctional drift in the M1–M2 range, where metabolic factors and scavenger receptors of these cells play a more significant role [75].

In a simplified form, inflammatory macrophages can be divided into four subsets, each of which collaborates with subpopulations of CD4^+^ T cells that are complementary to them in terms of proinflammatory function and mutual activation and, thus, form four immune response vectors (i1, i2, i3, and i-reg), each of which initiates progression of inflammation in a specific direction [119]. At the same time, multiple immune reactivity vectors can have functional overlap zones and be manifest in varying ratios in the inflammatory focus.

Vector i1: Th1 ↔ M1 (IL-1β, IL-6, IL-12, IL-15, IL-23, TNF-α, CCL-2-5, and CCL-8–11), as a rule, is realized with the participation of CD8^+^ cytotoxic T-lymphocytes and normal killer cells, being effective in infection with intracellular parasites, in antitumor immunity, in allograft rejection and delayed-type mononuclear hypersensitivity, and in some autoimmune processes. However, this response can damage own tissue.

Vector i2: Th2 ↔ M2a (IL-6, IL-10, TGF-β, CCL-17, CCL-22, and CCL-24) mutually restricts and competes with Vector i1. Eosinophils, basophils, and mast cells are actively involved in i2. It is most appropriate for metazoic infection and is involved in allergic inflammation, as well as in the processes of post-inflammatory regeneration and tissue repair, and in sensitive tissues (e.g., uterus), but it can contribute to fibrosis of internal organs and, in some cases, to tissue damage.

Vector i3: Th17 ↔ M2b (IL-1β, IL-6, TNF-α, and CCL-1) is functionally associated with neutrophils. Vector i3 is implicated in the formation of autoimmune inflammation, transplant rejection, and antitumor immunity and is less in conflict with other vectors for the development of productive inflammation. This vector can also be realized in extracellular bacterial and fungal infections.

Vector i-reg: Treg ↔ M2c vector (IL-10 and TGF-β) is realized in parallel with other variants of the immune response as a restrictive (suppressive) mechanism that prevents excessive inflammation and autoimmune aggression.

During inflammation, different immune vectors can be realized simultaneously or sequentially; hence, it is governed by the vector sum (superposition) of the immune system’s responses to local and systemic activities of multiple activation cues. This determines the dynamics of inflammation, its adaptation to the damage factor, and the complexity of its characterization in the in vivo system.

Chronic productive or mixed inflammation and its exacerbation—in the form of exudative–destructive inflammation, usually of the fibrinous type—are all symptoms of autoimmune diseases [126]. As previously stated, autoimmune inflammation is closely linked to vectors i1 and i3, with i-reg and, partly, i2 acting as a counterweight. Meanwhile, there is no doubt that the i2 vector is implicated in the development of not only fibrosis [127], but also tissue destruction [128] in various autoimmune disorders.

### 5.2. Typical Manifestations of Classical Inflammation in Nephrites

The main causes for the development of classic inflammation in the kidneys are autoimmune processes, infections, and kidney transplant rejection. The leading mechanism of autoimmune inflammation in the kidneys is the deposition of immune complexes in various compartments of the renal corpuscles, mainly in the renal glomeruli, with subsequent involvement of the rest of the kidney in autoimmune inflammation. As a rule, the development of glomerulonephritis predominantly proceeds according to the mixed type (fibrinous inflammation with the addition of productive inflammation). The main causes of this progressive pathology are the following [22,23,24,129,130,131]:deposition of soluble immune complexes (post-streptococcal, lupus, and mesangiocapillary nephritis, Schönlein–Henoch disease, and IgA nephropathy);antibodies to the basement membrane of the renal glomeruli (Goodpasture’s disease, also known as anti-glomerular basement membrane disease);antineutrophil cytoplasmic antibody (ANCA-associated vasculitis).

Complement activation, including the actions of complement anaphylaxins (C3a, C4a, and C5a) and membrane assault complex (C5b-C6-C7-C8-C9)n [132], is a significant pathogenetic trigger of the gradual replacement of functional nephrons by fibrous tissue in the abovementioned diseases. These effects are largely followed by the proinflammatory and chemotactic effect of complement on leukocytes, activation of the renal renin–angiotensin system, and involvement of cellular immunity factors in this process [132]. In these pathologies, complement is activated in a classical way with the deposition of factors C3b and C4b in the glomeruli [133,134]. In some autoimmune diseases (systemic lupus erythematosus), the processes of immunocomplex pathology become systemic, affecting many internal organs.

In relatively rare cases, the development of autoimmune glomerulonephritis may also be associated with alternative activation of complement, primarily in type 2 membranoproliferative glomerulonephritis (dense deposit disease—DDD) [135]. In DDD, substantial alternative complement activation with the creation of C3b, C3a, and C5a occurs not only in the kidneys but also at the systemic level, resulting in an extremely negative prognosis. That is why complement-blocking compounds may become promising drugs in the treatment of many nephropathies [136].

The Th1 and Th17 immune response mechanisms play a key role in the establishment of productive autoimmune inflammation in the kidneys, as they do in other autoimmune diseases [137,138,139]. Damage induced by Th17 manifests itself early, whereas glomerulonephritis caused by Th1 manifests itself later and is characterized by increased macrophage activation and differentiation in the direction of M1. As a result, some kinds of proliferative glomerulonephritis may be characterized by proinflammatory Th17 function, whereas others are characterized by Th1 function [137,140]. At the same time, the mechanisms of the i2 immune response are also involved in the development of renal inflammation, including the possibility of interstitial eosinophilic infiltration in DKD [95] and activation of mast cells [97]. The involvement of i2 cytokines (IL-4 and eotaxin) [141], other profibrogenic cytokines, and growth factors of T and mast cells [104] in the development of interstitial inflammation have also been noted. In DKD and nondiabetic nephropathies, renal inflammatory macrophages are characterized by separate properties of both M1 and M2 [142]. Thus, the development of immune inflammation in nephrites is primarily associated with the processes of complement activation and with the vectors of the immune response i1 and i3, but individual i2 factors can also be involved in the processes of renal fibrosis.

### 5.3. Possible Causes of the Transformation of Local ChLGI into Classic Inflammation in Diabetic Kidney Disease

The slow development of renal fibrosis up to the ESRD stage is due to a local ChLGI, without the hallmark morphological markers of classical inflammation, for example, in the case of the primary contracted kidney associated with essential hypertension [143]. In DKD, proinflammatory cell activation is accompanied by hypertrophy and damage to podocytes, their separation from the basement membrane, endotheliosis with the development of oxidative stress with hyperfiltration, and then glomerular sclerosis [14,144,145,146]. However, as previously mentioned, progressive ChLGI in DKD can acquire characteristic local signs of classical inflammation.

Usually, moderate chronic proinflammatory stress does not lead to the activation of inducible NO synthase (iNOS), but, on the contrary, promotes inhibition of the constitutive endothelial form of this enzyme (eNOS) [14]. In this case, the expression of iNOS in macrophages and endothelial cells is usually associated with the processes typical of classical inflammation, and at the systemic level, with shock states of systemic inflammation [52]. The phenomenon of reduced NO bioavailability at the systemic level, in turn, is associated with hypertension and some other systemic manifestations of ChLGI, and induction in iNOS cells requires overcoming a certain threshold in the intensity of tissue alteration and the level of development of “inflammatory microcirculation” (Figure 1). Reduced NO bioavailability is a common hallmark of many CKD types at the local level, and this disorder is almost universal in patients with end-stage renal disease [147], who have largely lost the renal parenchyma—the morphological substrate for classical inflammation. Several studies have demonstrated that higher glomerular eNOS expression and production of iNOS in inflammatory cells were associated with more severe DKD vascular problems and proteinuria [148]. Moreover, the cocultivation of podocytes and macrophages in a hyperglycemic environment led to the polarization of macrophages toward M1 with activation of iNOS and damage to podocytes [149]. Oxidative stress, in turn, contributes to the progression of DKD, and ROS accumulation is associated with the overproduction of proinflammatory cytokines, vascular inflammation, and fibrosis [150]. Thus, the activation of NF-κB in macrophages and their polarization towards M1 represent one of the key proinflammatory mechanisms in the kidney. However, in chronic inflammatory reactions such as diabetic kidney disease, M1 and M2 macrophages can coexist, resulting in persistent inflammation and fibrosis [151].

Negative dynamics of DKD is largely determined by the severity of proinflammatory stress in endotheliocytes and podocytes, including activation of the following signaling pathways:intercellular contact receptors Notch-1;membrane GTP-binding proteins;stress protein kinases MAPK (mitogen-activated protein kinase) and mTORC1 (mammalian target of rapamycin);many stress miRNAs and long non-coding (regulatory) RNAs;heat-shock proteins, especially HSP70;various proapoptotic factors;other factors of cellular stress, including chemokines, growth factors, and TGF-β1 [25,93,144,152].

Metabolic variables such as hyperglycemia, AGE buildup, and highly saturated FFA concentrations influence the negative dynamics of proinflammatory tissue stress, leading to damage to proximal tubule cells and glomerular podocytes in DKD [153]. All these are associated with the development of mitochondrial and oxidative stress and with the proinflammatory secretory phenotype of kidney cells [154,155]. Hypoxia is another important factor in the development of tissue stress in the kidneys, as it triggers a vicious cycle of capillary injury, inflammation, extracellular matrix deposition, and, eventually, fibrosis and nephron loss [27,103]. The cells of the proximal tubules, which make up 90% of the kidney cortex and are the main ATP-consuming cells in the kidneys (they spend most of ATP for active transport and reabsorption of solutes) and use the oxidative phosphorylation process for ATP synthesis only [156], are the most sensitive to hypoxia. Due to their high metabolic activity, tubular epithelial cells are particularly vulnerable to hypoxia and, after sustained damage, initiate an inflammatory response by recruiting inflammatory cells into the interstitium and secreting various profibrogenic cytokines such as PDGF, TNF-α, and IL-6, which not only activate fibroblasts but also contribute to the epithelial–mesenchymal transition. The involvement of T lymphocytes, other leukocytes, and exudative–fibrinous reactions in the process, which are already signs of classical inflammation, may increase with the escalation of the above-mentioned proinflammatory mechanisms [94,96,97]. Perhaps there is a transitional (“gray”) zone between local ChLGI and classical inflammation in such cases, for example, in foci of atherosclerosis of big arteries. In such cases, there is evidence of both local ChLGI and productive inflammation without any microcirculatory component [52].

Increased migration of type 2 diabetes mellitus T cells (including CD8^+^ T effectors) into adipose tissue is not a rare event [157]. In this scenario, however, there is no need to discuss traditional adipose tissue inflammation. This necessitates a greater activation of microvessels, the action of exudation, and more extensive leukocyte migration. The level of proinflammatory tissue stress in the kidneys in diabetes mellitus may be higher than in most other tissues for a variety of reasons that are not fully understood [157,158]. The high density of the microvascular network, a high degree of metabolism and metabolic activities in the renal tissue, and a high local concentration of mesangial macrophages may be explanations for this trait. A shift of T-helpers toward Th1 and Th17 is seen in DKD (type 2 diabetes mellitus), and the concentration of these cells and their cytokines (IL-17, IFN-γ, and TNF-α) in the blood corresponds with albumin levels in urine and blood creatinine [159]. At the same time, metabolic non-antigenspecific dysfunctions, such as hyperglycemia factor, AGE accumulation in the blood, and lipotoxicity factors, may be associated with T-cell and macrophage activation and polarization in the direction of i1 and i3 responses, which are common in type 2 diabetes mellitus [157].

### 5.4. Typical Features of Inflammation in Dysfunction and Rejection of the Renal Allograft

Kidney transplantation is presently a radical treatment for individuals with ESRD, and it is associated with a lower mortality rate and improved quality of life when compared to dialysis [160]. After PH rejection, the levels of uremic toxins in the recipient’s blood drop, as do the levels of active endotoxin—LPS (improved intestinal barrier function), a soluble form of its coreceptor—sCD14 (presepsin), and blood levels of proinflammatory cytokines [161,162]. However, in some cases, graft dysfunction develops despite the use of immunosuppressive therapy. The local and systemic proinflammatory status of the recipient worsens as renal graft failure develops. Simultaneously, prominent pathomorphological signs of most variations of this dysfunction, according to the Banff classification, reflect the involvement of molecular mechanisms of tissue stress in these processes, resulting in fibrinous and productive inflammation [3,161,162,163].

Chronic glomerulonephritis is the leading cause of ESRD in 25% of dialysis patients and 45% of kidney transplant recipients [164]. At the same time, recurrent glomerulonephritis contributes insignificantly to graft loss, but when graft survival improves, the impact of recurrent glomerulonephritis on the graft’s fate becomes increasingly important [164]. Idiopathic nephrotic syndrome, which is resistant to corticosteroids, recurs in 30–50% of transplant recipients [165]. In this case, the recurrence of nephrotic syndrome may be associated with Th2 factors [165].

Chronic allograft dysfunction (CAD) is caused by a variety of factors, including drug nephrotoxicity, infections, systemic metabolic and microcirculatory disorders in patients with ESRD, and the effects of T lymphocytes on immune rejection [166]. Another aspect of chronic allograft nephropathy is that, in addition to macrophages and T lymphocytes, the number of mast cells in the graft usually increases [167].

The main cause of acute graft dysfunction is an immunological conflict—acute graft rejection [168]. During graft rejection, the complement system plays an important role in the development of numerous forms of graft inflammation [169]. In the presence of an immune conflict, it is more common to participate in the rejection of T cells with Th1 features [160,170]. T cells with Th1 features, Th17 cells, and the deficiency of the anti-inflammatory function of Treg contribute to the immune mechanisms of kidney rejection [171,172]. Moreover, B cells also participate in the development of this immune conflict due to the secretion of chemokines GRO-α (CXCL1), RANTES (CCL5), and MCP-1 (CCL2), which contribute to the development of productive inflammation [173]. High levels of alloantibodies to HLA antigens (especially HLA class II) and markers of allograft endothelial cell activation were also shown to be predictive for acute and chronic rejection [171].

Ischemic reperfusion injury in kidney transplantation is the main cause of delayed graft function, an event associated with an increased risk of acute graft rejection. In this case, the accumulation of Th1 inside the graft is noted, with Th17 accumulated to a lesser and Th2 to an even lesser extent [174].

Thus, renal transplant rejection and many variants of its dysfunction are based on mechanisms of exudative–fibrinous and productive inflammation associated with complement-dependent antibody responses, as well as vectors of cell-mediated responses—i1 and i3 (however, i2 response pathways in the progression of renal fibrosis cannot be ruled out).

## 6. Systemic Inflammatory Processes in ESRD

### 6.1. General Characteristics of Systemic Inflammation

SIR is an obvious sign of system change and can be defined as the accumulation of tissue proinflammatory stress products in the blood. The development of SIR can be a companion to various processes, including systemic ChLGI, classical inflammation (a generalization of inflammatory mediators from the inflammatory focus), and chronic systemic inflammation itself (ChSI) as a general pathological process. Acute systemic inflammation is associated with the phenomenon of systemic inflammatory microcirculation and life-critical microcirculatory disorders (MDs) [175]. Various resuscitation syndromes with formal protocols reflect acute systemic inflammation clinically, making it easier to formalize the verification criteria for this general pathological process [175,176]. However, in chronic pathologies, MDs proceed latently. This makes the task of separating ChSI from other SIR variants more difficult, even considering the severity of the general condition of ESRD patients. Excessive microvessel permeability, postcapillary microthrombosis, dilatation of arterioles, and capillary sphincters are associated with acute tissue perfusion disorders, while ChSI is associated with a gradual decrease in the density of the microvascular network vasoconstriction of arterioles, often complicated by hypertension, atherosclerotic arterial stenoses, and major arterial stenoses [37]. In general, ChSI features are not generated discretely; hence, their verification necessitates probabilistic multivalued logic approaches that allow intermediate values between “true” and “false.” When compared to acute systemic inflammation, ChSI can be thought of as a prolonged stage of pre-systemic inflammation (Figure 1).

### 6.2. Features of SIR Development in ESRD

Currently, several SIR markers can be identified, the concentration of which in the blood is strongly linked to the dynamics of chronic renal failure in general. These factors include accumulation in the blood [177,178,179,180,181,182,183,184,185] of the following:various ROS;proinflammatory cytokines: TNF-α, IL-6, and IL-18;chemokines: IL-8 (CXCL8), IL-34, SDF1α (CXCL12), MCP-1 (CCL2), and MIP-1β (CCL4);growth factors: GM-CSF (granulocyte-macrophage colony-stimulating factor), FGF-23, and HGF (hepatocyte growth factor);soluble forms of receptors: sTNFR1 and sTNFR2, sCD40L, and sCD163 (SR-I3);cyclophilin A as an inducer of proinflammatory cytokines [186].

ANGPTL2 (angiopoietin-like protein 2), endocan, thrombomodulin, and soluble versions of vascular endothelial growth factor 1 and 2 receptors (sVEGFR1/2) are all markers of endothelial dysfunction [177,187,188,189]. SIR factors have been implicated in the etiology of ESRD in hemodialysis patients [190] and in individuals with chronic allograft dysfunction [161]. Simultaneously, the severity of SIR in hemodialysis patients did not appear to be affected by the original disease—DKD, chronic primary glomerulonephritis, or pyelonephritis [191]. Our studies of blood plasma in ESRD patients revealed the following [192]:multiple increases in average and median values of TNF-α, IL-8 and the soluble form of the IL-2 receptor (sCD25);less significant (approximately two-fold from the upper level of the norm) changes in C-reactive protein (CRP), IL-6, and TGF-β1;at the same time, the concentration of anti-inflammatory IL-10 did not differ significantly from its level in the blood of conventionally healthy people.

A separate area of SIR assessment in ESRD is the determination of extracellular vesicles (EV) in the blood [193,194]. EVs are formed in various cells and platelets, including exosomes of intracellular origin (<0.1 μm) and exosomes or microvesicles separated from the plasma membrane (~0.1–1 μm). The role of EVs, especially exosomes, is intercellular communication by transferring regulatory proteins, mRNA, and miRNA into recipient cells as target nanovectors. EVs can mediate crosslinking between different types of cells in the kidney to maintain tissue homeostasis, as well as between the kidneys and other organs in physiological and pathological conditions. In the latter case, EVs are associated with tissue stress and act as SIR factors. Moreover, EVs can both enhance blood coagulation due to their powerful procoagulant effects and suppress platelet function; thus, the evaluation of EV can be useful for predicting the complications of ESRD [193,194,195].

### 6.3. Systemic Inflammatory Phenomena Specific to ESRD

As previously stated, there are several reasons for the development of the phenomenon of systemic damage—a trigger for the development of ChSI—with distinct forms of ESRD development. In general, this phenomenon is less pronounced in chronic illnesses than in acute critical conditions. However, even with ESRD, there is a constant and considerable increase in myoglobin levels in the blood [192], as well as myocardial-specific troponins I and T in some patients [161,190,196]. Meanwhile, with ESRD, determining indicator aminotransferases appears to be less useful for confirming systemic diseases [197]. The development of sarcopenia and cachexia in ESRD is both a consequence and a cause of systemic diseases [37,198,199]. These diseases are caused by interconnected anomalies in skeletal muscle, liver, and adipose tissue metabolic activities, and they are characterized by disturbances in metabolic cycles and accumulation of lipotoxicity factors in the blood [52].

The key attribute of any type of systemic inflammation as a general pathological process is the phenomenon of systemic inflammatory microcirculation. During ESRD, according to laser speckle flowgraphy with adenosine loading and other methods for assessing MD, significant weakening of the microcirculatory reserve in integumentary tissues and internal organs is revealed, which correlates with an increase in creatinine blood levels and the degree of decrease in GFR [37,200,201,202]. Disorders of microvascular (endothelial) function, in turn, play a significant role in the progression of cardiovascular and renal diseases [203,204,205]. According to sublingual lateral dark-field microscopy, the degree of MD in patients with ESRD can significantly decrease after a successful kidney transplant [205]. However, the disturbance (impairment) of microvascular blood flow up to a certain point may not be reflected by the changes in macrohemodynamics [204].

The development of microthrombus formation (particularly intense in postcapillaries) or, otherwise, disseminated intravascular coagulation (DIC) is one of the signs of systemic MD. DIC symptoms are usually latent in chronic MD patients and do not fit the criteria for DIC resuscitation syndrome. Changes in the hemostatic system during uremia, on the other hand, can play a significant role in vascular complications, manifesting as increased levels of fibrinogen (acute phase protein) in the blood and activation of DIC markers, such as soluble fibrin monomeric complexes and D-dimers [206,207]. Meanwhile, prothrombotic alterations can be seen in ESRD patients even when secondary thrombocytopathy is present, which is one of the causes of hemorrhagic diathesis [208]. As a result, monitoring hemostatic alterations in hemodialysis patients is critical for the application of appropriate anticoagulant therapy regimens on time.

Intravascular para-coagulation and blood contact with a foreign surface during PH conduction promote complement activation and the accumulation of anaphylatoxins C3a and C5a in the blood, which additionally contributes to the activation of intravascular leukocytes, endothelial cells, and perivascular mast cells [209,210,211].

Perivascular mast cells are direct participants in the inflammatory reactions of microvessels. Severe pruritus is one of the symptoms of ESRD that affects the quality of life and is associated with poor outcomes in hemodialysis patients [212]. Proliferation and activation of mast cells in the dermis may be one of the causes of pruritus [212,213]. In addition, uremic pruritus in hemodialysis patients is associated with an increase in the level of i2 response cytokines in the blood (IL-13 and IL-31) linked to mast cell activity [214]. At the same time, after kidney transplantation, the increased number of mast cells in tissues in individuals with ESRD can return to normal [215]. The level of tryptase in the blood reflects the degree of systemic pathogenic activation of mast cells in ESRD [216]. In ESRD, blood levels of tryptase, TGF-β, and IL-6 are positively correlated with the degree of venous intimal hyperplasia and the generation of venous neointimal hyperplasia [217]. It has been observed that IL-9 production and mast-cell degranulation promote neointimal hyperplasia and prevent re-endothelialization of a vein transplanted into an animal in an advanced stage CKD model (mice with 5/6ths nephrectomy) [218]. These findings imply that IL-9 and mast cells may play a role in the pathophysiology of fistula insufficiency caused by neointimal venous hyperplasia.

A significant risk factor for cardiovascular complications in ESRD is the systemic activation of innate immunity factors: complement, phagocytic leukocytes, and macrophages [219]. In general, ESRD is characterized by the dysregulatory activation and dysfunction of the mononuclear phagocyte system, including blood monocytes and dendritic cells involved in immunocompetent organs, which is one of the factors not only for changes in the vascular system but also for dysfunction of the immune system [220]. Pathological activation of neutrophils with the accumulation of tissue-damaging proteinases, cationic proteins, and ROS in the blood is also essential in the development of systemic intravascular inflammation [221]. In addition, the formation of neutrophil extracellular traps (NETs) in the intravascular environment is a mechanism of tissue damage and an obvious risk factor for cardiovascular complications in patients with ESRD [222]. This variant of programmed necrosis of neutrophils and other phagocytes is designated as NETosis. In particular, during a hemodialysis session, neutrophil NETosis is accompanied by the release of myeloperoxidase, elastase, and other phlogogenic factors into the bloodstream [223,224]. Furthermore, in ESRD, intravascular eosinophil activation is observed, as evidenced by a rise in blood eosinophilic cationic protein (ECP), a factor in tissue change [192,225].

The distress reaction of the hypothalamic–pituitary–adrenal system is a universal manifestation of many severe human pathologies, but it is especially pronounced in the development of systemic inflammation. Patients with ESRD experience significant increases in plasma cortisol levels [192,226,227]. Long-term hemodialysis treatment, on the other hand, can reveal signs of adrenal insufficiency, including arterial hypotension [228] and, in some circumstances, even mimicking Addison’s disease [229].

Thus, ESRD at the system level reveals a range of not only quantitative but also qualitative changes that go beyond the concept of systemic ChLGI, which rather correspond to another general pathological process—ChSI.

### 6.4. Use of the Integral Criterion of Systemic Inflammation in ChSI

Currently, the theory of ChSI as an independent type of general pathological process is incomplete. ChSI is not a commonly accepted definition. However, the available data make it possible to substantiate the necessity and possibility of conducting a preliminary intergroup analysis using the ChSI integral verification methods. At the same time, a special problem of ChSI verification is the need for a more fundamental qualitative/semiquantitative SIR assessment.

A common problem of SIR indicators is nonlinearity, randomness, and a low degree of correlation among them regarding their changes in blood, which predetermine the use of integral indicators for a stable SIR characteristic. For the integral assessment of SIR, it is necessary to use the most informative markers, and the information content of the integral scale in terms of specificity and sensitivity for each situation should be higher than the use of separate tests. In addition, the integral criteria should differentiate, according to the severity of SIR, various typical inflammatory processes underlying the pathogenesis of ESRD. For these purposes, we use a universal open (with the possibility of replacing indicators) reactivity level (RL) scale, which was originally developed to predict acute critical complications in sepsis and trauma [175,176], as well as assess SIR in chronic diseases [161,190]. The RL scale includes six semiquantitative ranges of the distribution of SIR concentrations in the blood (0–5, indicated as scale points):RL-0 characterizes the reference values of the norm;RL-1 is typical of ChLGI and SIR manifestations in classic acute and chronic inflammation;RL-2 is typical of severe acute purulent-inflammatory processes of the classical type;RL-3 is the overlap zone SIR of classical acute purulent inflammation and systemic inflammation;RL-4 and RL-5 are levels typical of the hyperergic variant of acute systemic inflammation;RL-3–5 in chronic processes, in our opinion, a priori confirm ChSI, while verification of ChSI at lower SIR values (RL-1–2) requires additional criteria.

In most cases, we use five indicators to calculate RL (CRP, IL-6, IL-8, IL-10, and TNFα); for each of them, we set the corresponding ranges of the pathogenetic significance of their concentrations in the blood. Furthermore, for each patient, three indicators with the highest individual RL values are identified and averaged to calculate the RL scale. RL values (0–5 points) are used to calculate the more integral ChSI scale, which takes into account, in addition to SIR levels, the presence of some other ChSI phenomena.

The ChSI integral scale [161] is an analogue of the scale for assessing systemic inflammation in acute critical conditions adapted to chronic pathologies [175]. The ChSI scale takes into account the RL and other phenomena (1 point for the presence of each). Its values range from 0–8 points (Table 1).

The presence of ≥3 points on the ChSI scale with RL >0 indicates the possible presence of ChSI.

The frequency of detection of integral signs of ChSI and a separate criterion for para-coagulation (D-dimer >500 ng/mL) as a key particular phenomenon of ChSI were compared between groups of patients with ESRD (who received PH replacement therapy and had chronic renal transplant dysfunction) and other chronic pathologies (Table 2). The causes of ESRD in patients receiving PH were primary chronic glomerulonephritis, diabetes mellitus, and chronic pyelonephritis, and the cause of renal disease in all patients prior to kidney transplantation was chronic glomerulonephritis. The detailed statistics from the Table 1 featured in our other publications [161,192,230]. According to the data obtained, the following pathologies were associated with a high risk of developing ChSI (20% according to the ChSI scale criterion):systemic autoimmune diseases (systemic lupus erythematosus, rheumatoid and reactive arthritis, and primary antiphospholipid syndrome);critical atherosclerotic ischemia of the lower extremities, complicated by gangrene of the toes;programmed hemodialysis (glomerulonephritis, diabetes mellitus, and chronic pyelonephritis);chronic allograft dysfunction regardless of the morphological variant of Banff classification based on biopsy data (taking into account the presence of large-scale anti-inflammatory and immunosuppressive therapy).

**Table 2 ijms-22-11453-t002:** The frequency of the values of reactivity level (RL), D-dimer >500 ng/mL (D-d), and ChSI Score (ChSI ≥3 points) in chronic diseases (%).

Group	n	RL						D-d	ChSI
		0	1	2	3	4	5		
Healthy people, age <55 years old	50	0	0	0	0	0	0	0	0
Conditionally healthy people, age >65 years old	18	88.9	11.1	0	0	0	0	0	0
PID	16	75	25	0	0	0	0	0	0
Chronic phlegmons	42	19	78.6	2.4	0	0	0	9.5	7.1
Hypertension, PMS	16	93.7	6.3	0	0	0	0	0	0
Elderly patients (>65 years old) with CHF and encephalopathy	49	53.1	36.7	10.2	0	0	0	32.7	2.0
Atherosclerotic stenosis of CFA with gangrene	38	5.3	31.6	52.6	10.5	0	0	47.4	57.9
AIT	29	79.3	20.7	0	0	0	0	0	0
PsA	12	33.3	50	16.7	0	0	0	8.3	8.3
Ankylosing spondylitis	27	44.5	33.3	22.2	0	0	0	11.1	11.1
Valvular heart diseases	15	53.5	33.3	13.3	0	0	0	13.3	13.3
ReA	30	46.7	33.3	20	0	0	0	23.3	20
Rheumatoid arthritis	42	31	47.6	19	2.4	0	0	54.8	38.5
SLE	49	8.2	4.1	16.3	32.6	34.7	4.1	40.8	75.5
PAPS ^1^	5	0	0	20	80	0	0	100	100
End-stage renal disease (program hemodialysis) ^2^	42	4.8	16.6	54.8	21.4	2.4	0	38.1	88.1
CAD ^3^	23	8.7	69.6	17.4	4.3	0	0	21.7	43.5
Normal function of renal allograft	24	58.3	25	16.7	0	0	0	4.2	0

**Note**: PID—Pelvic inflammatory disease; PMS—premenstrual syndrome; CFA—common femoral artery; AIT—autoimmune thyroiditis; PsA—psoriatic arthritis; ReA—reactive arthritis; SLE—systemic lupus erythematosus; PAPS—primary antiphospholipid syndrome; CAD—chronic allograft dysfunction; ^1^ in women with recurrent miscarriage; ^2^ initial diseases: chronic glomerulonephritis, diabetes mellitus, and chronic pyelonephritis; ^3^ there are no correlations between the values of RL and CSI with morphological variants of renal allograft pathology.

RL-2–3 and signs of para-coagulability (D-dimer >500 ng/mL) are common characteristics of ChSI. In two individuals with systemic lupus erythematosus, we found an RL-5 level of SIR that was uncommon for chronic disease (a rise in proinflammatory cytokines in the blood by thousands and tens of thousands of times). Furthermore, a meta-analysis found that almost half of patients with systemic lupus erythematosus develop lupus nephritis, which is associated with a 30% risk of ESRD [231]. As a result, some ESRD patients have RL-4, which is also associated with acute systemic inflammation. Thus, it can be concluded that a high level of SIR and the severity of other systemic disorders, which are atypical for ChLGI, lead to the development of ChSI in most patients with ESRD.

## 7. Conclusions

Any pathological process is unique, which determines the use of personalized approaches in medical practice. However, these approaches must take into account the fundamental laws, as well as the separation of the specific and the general, where the role of the general is played by general pathological processes. The most fundamental general pathological processes associated with the molecular mechanisms of inflammation are ChLGI and classical and systemic inflammation. The overwhelming majority of human diseases are based on certain combinations of these basic processes. In particular, we were previously able to show the involvement of all these processes in the pathogenesis of COVID-19 and its critical complications [232]. In our opinion, the involvement of these three processes is also evident in the pathogenesis of CKD and ESRD. Moreover, the most obvious role of ChLGI is observed in DKD (Figure 3).

Primary and secondary glomerulonephritis (usually in autoimmune diseases) are characterized by classical inflammation of the fibrinous–productive type in the kidneys. Certain autoimmune diseases, such as systemic lupus erythematosus and dense deposit disease, may be associated with a chronic variant of systemic inflammation independently of ESRD. However, in most other cases, ESRD is likely to play a defining role in the development of ChSI. At the same time, kidney transplantation and accompanying immunosuppressive therapy largely relieve the manifestations of ChSI. However, chronic allograft dysfunction may be accompanied by a recurrence of ChSI. In general, as CKD progresses, the morphological platform for the development of local inflammatory processes decreases due to the replacement of the renal parenchyma with weakly vascularized fibrous tissue.

Meanwhile, with ESRD, the pathogenetic role of systemic manifestations of inflammatory processes increases the development of secondary pathologies on the part of the cardiovascular system and other organs with the formation of a vicious pathogenetic cycle. These patterns can be taken into consideration for the development of typical and individual approaches for patient treatment. One positive example in the implementation of this approach is the multicenter Nephropathy in Diabetes Type 2 (NID-2) study, which demonstrates that a strategy based on intensive treatment of major risk factors and systemic effects on key links in pathogenesis is more effective than standard care in the prevention of cardiovascular complications in type-2 diabetic kidney disease [233].

## Figures and Tables

**Figure 1 ijms-22-11453-f001:**
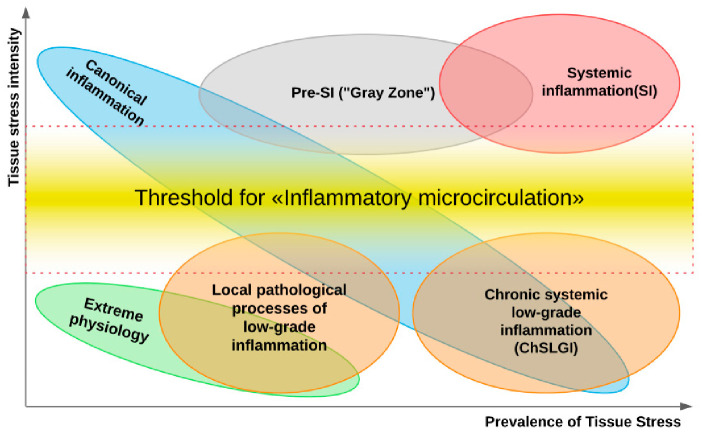
Tissue stress and general pathological processes (from Gusev E. et al., 2021). Note: The ratio of intensity to prevalence of damaging factors initiating a “response” in the form of tissue proinflammatory stress—a common pathogenetic underpinning of all pathological processes—can be used to distinguish three “big” general pathological processes (classical inflammation, SI, and ChLGI). The “inflammatory microcirculation”—a reaction of microvessels that causes a clinically significant exudative reaction and enormous migration of leukocytes into a damaged area with the emergence of an inflammation focus—is the key differentiating phenomenon for their distinction. (1) The presence of an inflammation focus (upper-left corner in Figure 1) and, in some cases, systemic manifestations of tissue stress at a subthreshold level for “inflammatory microcirculation” characterize classical inflammation. These systemic reactions are, for the most part, protective and aimed at providing resource support for the inflammation’s focal point (immune response, acute-phase response of the liver, forced leukocytopoesis, neuroendocrine stress, and mobilization of metabolic reserves). (2) Systemic inflammation (SI) is characterized by shockogenic systemic “inflammatory microcirculation” and presence of a transitional “gray” zone, in which it is impossible to prove or refute the presence of SI in situations of “pushing” variant development (upper right-hand corner of the figure). (3) Para-inflammation (low-intensity inflammation) can be local (see the figure in the lower left-hand corner of figure) or can spread throughout the body (see ChLGI) (lower right-hand corner of figure) Signs of endotheliosis and, in certain circumstances, latent microcirculatory disorders may be encountered within these types of tissue proinflammatory stress.

**Figure 2 ijms-22-11453-f002:**
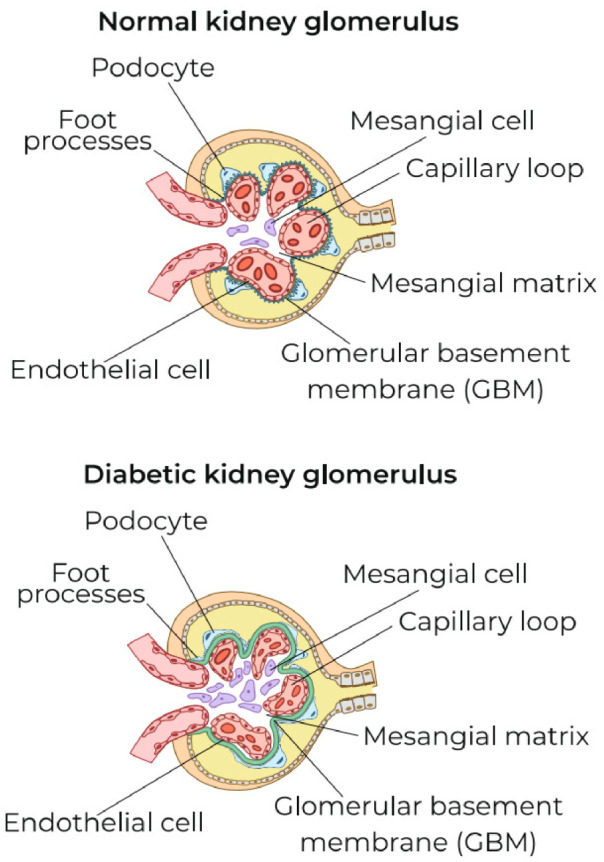
Characteristic features of local low-cell inflammation in the renal corpuscles in diabetic kidney disease. Note: The typical initial changes in the renal corpuscles in diabetic kidney disease are the activation and proliferation of mesangial macrophages, thickening of GBM, a decrease in the number of podocytes, and the density of their processes. However, signs of canonical inflammation have not yet been expressed: there is no exudation (with deposition of fibrin and complement factors and significant migration of leukocytes into the intercapillary space of the renal glomeruli and the cavity of Bowman’s capsule.

**Figure 3 ijms-22-11453-f003:**
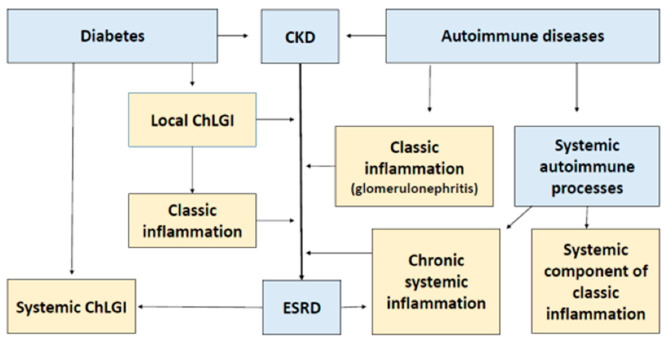
Relationship among the root causes of chronic kidney disease (CKD), end-stage renal disease (ESRD), and general pathological processes. On the left side of the scheme, diabetes mellitus and hypertension (against the background of obesity and metabolic syndrome) contribute to the emergence of a systemic proinflammatory status of the chronic low-grade inflammation (ChLGI) type. Simultaneously, pronounced local manifestations of ChLGI contribute to the development of CKD. In DKD, ChLGI is frequently transformed into alterations that resemble the classical variants of inflammation. These processes also lead to ESRD. ESRD, in turn, is an obvious inducer of changes in the proinflammatory status of the body according to the variant of chronic systemic inflammation (ChSI). On the right side of the diagram, many autoimmune diseases are associated with primary and secondary chronic glomerulonephritis, characterized by fibrinous productive variants of canonical inflammation and CKD, which can progress to ESRD. In some systemic autoimmune diseases, especially in systemic lupus erythematosus, secondary manifestations of chronic glomerulonephritis and primary (before ESRD) development of ChSI are noted, which additionally contribute to the progression of CKD to ESRD with the formation of a vicious pathogenetic cycle.

**Table 1 ijms-22-11453-t001:** The Chronic Systemic Inflammation (ChSI) scale.

ChSI Phenomena	Partial ChSI Criteria	Unit	Norm	ChSI Scale Points
Systemic inflammatory response	RL scale	Points (0 to 5)	0	1 RL point = 1 ChSI scale point
Microthrombus formation	D-dimers >500	ng/ml	<250	1 point
Systemic alteration	Myoglobin >60	ng/ml	<25	1 point
	Troponin I >0.2	ng/ml	<0.2	
Distress reaction of the hypothalamic pituitary adrenal system	Cortisol >690	nmol/L	138–690	1 point
	Cortisol <100			

## Data Availability

Not applicable.

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
