# Peer review of "The Pathogenesis of End-Stage Renal Disease from the Standpoint of the Theory of General Pathological Processes of Inflammation"

_ijms, 2021, doi:10.3390/ijms222111453_

Round 1

Reviewer 1 Report

The manuscript is interesting. The topic, even though it has been addressed in other reviews, remains hot for the reader. The references are quite up-to-date.

This reviewer raises some issues that authors have to address.

1- Paragraph 5.3. “Possible reasons for the transformation of local ChLGI into classic inflammation in renal disease” focuses mainly on the inflammatory manifestations of diabetic nephropathy. It would therefore be appropriate to rename the paragraph appropriately, and structure it accordingly.

 2- Actually, the term "diabetic nephropathy" has been replaced from some years by "diabetic kidney disease" (DKD). Please, rename this kidney complication of diabetes correctly.

 3- In Conclusion the authors state “Any pathological process is unique, which determines the use of personalized approaches in medical practice” and “…with ESRD, the pathogenetic role of systemic manifestations of inflammatory processes increases the development of secondary pathologies on the part of the cardiovascular system and other organs with the formation of a vicious pathogenetic circle. These patterns shall be taken into consideration for the development of typical and individual approaches for patient treatment.” Actually,primary and secondary prevention of morbidity and mortality (especially cardiovascular) is at the center of modern therapy for DKD. Therefore, it is appropriate to underline that very recently this goal has been reached for the first time by randomized multicenter NID-2 study through a multifactorial intervention in albuminuric type 2 diabetic subjects, without previous MACE (Cardiovasc Diabetol (2021) 20:145. doi: 10.1186/s12933-021-01343-1). From my point of view, this issue should be adequately commented in conclusion section.

 4- In figure 1, the abscissa and ordinate values should be inserted in the same figure and not only in the notes. Furthermore, the reading of this figure is not easy, nor is it immediate to understand. Can it be made clearer for the reader?

 5- The manuscript needs a linguistic revision of a native English speaker.

Author Response

Dear Mr. Reviewer!

We thank you for reviewing our manuscript. Your comments are of great value to us, as they show that some terms need to be corrected and the wording made clearer for readers. We agree with your comments and have made the following corrections to the manuscript.

Reviewer 2 Report

  1. This review is acceptable. The authors provide a systemic review for the inflammatory regulation of pathological processes from chronic kidney disease to end-stage chronic renal disease (ESRD). It should be informative and useful for readers of this field.
  2. Only two figures and one table are not easily to help readers to quickly realize the complicated regulation of the inflammatory regulation of this disease process.
  3. A few typographic mistakes might be revised before submission.

Author Response

Dear Mr. Reviewer!

We thank you for reviewing and appreciating our manuscript. Your comments are of great value to us, as they show us how to make the article more understandable for readers.

Round 2

Reviewer 1 Report

No further comments.